# Atomically resolved phase transition of fullerene cations solvated in helium droplets

M. Kuhn[1], M. Renzler[1], J. Postler[1], S. Ralser[1], S. Spieler[1], M. Simpson[1], H. Linnartz[2], A.G.G.M. Tielens[2], J. Cami[3,4], A. Mauracher[1], Y. Wang[5,6,7], M. Alcamí[5,6,7], F. Martín[5,6,8], M.K. Beyer[1], R. Wester[1], A. Lindinger[9] & P. Scheier[1]

Helium has a unique phase diagram and below 25 bar it does not form a solid even at the lowest temperatures. Electrostriction leads to the formation of a solid layer of helium around charged impurities at much lower pressures in liquid and superfluid helium. These so-called 'Atkins snowballs' have been investigated for several simple ions. Here we form $He_nC_{60}^+$ complexes with $n$ exceeding 100 via electron ionization of helium nanodroplets doped with $C_{60}$. Photofragmentation of these complexes is measured by merging a tunable narrow-bandwidth laser beam with the ions. A switch from red- to blueshift of the absorption frequency of $He_nC_{60}^+$ on addition of He atoms at $n = 32$ is associated with a phase transition in the attached helium layer from solid to partly liquid (melting of the Atkins snowball). Elaborate molecular dynamics simulations using a realistic force field and including quantum effects support this interpretation.

[1] Institut für Ionenphysik und Angewandte Physik, Universität Innsbruck, Technikerstrasse 25, Innsbruck A-6020, Austria. [2] Leiden Observatory, University of Leiden, P.O. Box 9513, NL-2300 RA Leiden, The Netherlands. [3] Department of Physics and Astronomy/Centre for Planetary Science and Exploration (CPSX), The University of Western Ontario, London, Ontario, Canada N6A 3K7. [4] SETI Institute, 189 Bernardo Avenue, Suite 100, Mountain View, California 94043, USA. [5] Departamento de Química, Módulo 13, Universidad Autónoma de Madrid, 28049 Madrid, Spain. [6] Instituto Madrileño de Estudios Avanzados en Nanociencia (IMDEA-Nanociencia), Cantoblanco, 28049 Madrid, Spain. [7] Institute for Advanced Research in Chemical Sciences (IAdChem), Universidad Autónoma de Madrid, 28049 Madrid, Spain. [8] Condensed Matter Physics Center (IFIMAC), Universidad Autónoma de Madrid, 28049 Madrid, Spain. [9] Institut für Experimentalphysik, Freie Universität Berlin, Arnimallee 14, 14195 Berlin, Germany. Correspondence and requests for materials should be addressed to P.S. (email: paul.scheier@uibk.ac.at).

Tagging of ions with rare gas atoms and in particular with helium (He) provides an elegant method to measure absorption spectra of cold ionic species with minimum disturbing effects of a matrix[1–4]. Very recently, several laboratories developed instruments to master He tagging of complex molecules, which requires cooling of the ions in cryogenic traps[5–9] or supersonic jets[10]. An alternative method to form ions with He attached is the ionization of doped helium droplets[11,12] or pickup of ions by neutral helium droplets[13,14]. By choosing appropriate conditions, the number of attached He atoms can range from a few to several million atoms. Here we take advantage of this property and study the solvation of $C_{60}^+$ with helium, from the single atom limit to beyond the completion of several layers, by using messenger-type spectroscopy[4–9] and molecular dynamics (MD) simulations. We show the appearance of distinct changes in the matrix shift reflecting phase transitions of the adsorbed helium from solid to liquid and from liquid to superfluid. The changes manifest in $C_{60}^+$ absorption line positions recently assigned to several diffuse interstellar bands (DIBs)[9,15,16].

The formation of a solid layer of helium around an ionic impurity in bulk superfluid helium is often referred to as 'Atkins snowball', and has been investigated both experimentally and theoretically for several simple ions[17–21]. Gas-phase experiments of helium droplets containing an ion offer the opportunity to study the structure of these snowballs in detail. Around an ionic core, helium atoms form a solvation shell with a characteristic number of atoms. The size of this shell can be determined from distinctive steps in the dissociation energy and thus of steps in the ion abundance measured in mass spectrometers. The size may be 12 helium atoms for small cations, such as $Ar^+$ (refs 22,23) and up to 60 or 62 atoms for $C_{60}^+$ or $C_{70}^+$ fullerene cations, respectively[24,25]. Although these so-called magic number effects are well known, their relation to Atkins snowballs and the onset and extent of superfluidity remains obscured.

For a growing number of ad-atoms, it is not only the interaction with the surface that is important but also the mutual interaction between the solvent atoms. Fullerene cations, such as $C_{60}^+$, provide particularly powerful probes of the transition from adsorbant interaction with the host cation to mutual adsorbant interaction, as—in contrast to planar aromatic structures—the curved surface allows a fully covered commensurate helium monolayer[24,25]. Here we use this aspect, in conjunction with the characteristic wavelength shift (typically around 0.02 nm for the first adsorbed He atom)[9] introduced into electronic transitions by the He-$C_{60}^+$ interaction, to follow the transition from the solid to the liquid phase as a function of the number of helium atoms adsorbed. These measurements are performed by recording the wavelength-dependent changes in mass signal for different cluster sizes, $He_nC_{60}^+$.

## Results

**Mass spectra**. Figure 1 shows one mass spectrum off-resonant with the $He_nC_{60}^+$ ions at 962.21 nm compared with three mass spectra for laser wavelengths between 964.55 and 965.65 nm, close to the bare $C_{60}^+$ electronic excitation around 964 nm. The red circles indicate the peak of $He_n - {}^{12}C_{60}^+$ determined via a fitting routine[26]. Supplementary Fig. 1 shows the detailed result of the analysis of the mass spectrum for the ion $He_{17}C_{60}^+$. A movie showing the changes in a section of the mass spectrum throughout the scanned wavelengths is available as Supplementary Movie 1. Different parts of the mass spectrum are depleted, depending on the laser wavelength. This depletion can be regarded as hole burning of the cluster ion signals by the laser. For 964.55 nm, clusters around $n = 15$ are depleted. At longer wavelengths around 965 nm, two regions of the mass spectrum are diminished, which correspond to ~20 and 55 physisorbed helium atoms. For 965.65 nm in particular, low-ion signals are observed at $n = 30$ and 34.

**Absorption spectra**. Figure 2 shows seven representative absorption spectra for 2, 3, 6 and 32, 38, 39, 40 He atoms attached to $C_{60}^+$ at the $C_{60}^+$ electronic transitions near 959 and 964 nm, respectively. The centre positions of the absorption spectra for the bare $C_{60}^+$ ion corrected to vacuum are indicated by vertical sticks and taken from Campbell et al.[9]. The absorption strongly depletes the ion yield to minima at different wavelength positions with linewidths of ~0.2–0.6 nm, full width at half maximum. Absorption spectra for a wider range of attached He atoms for the electronic transition near 964 nm are shown in Supplementary Fig. 2.

The resulting line centre positions for the absorption spectra of $He_nC_{60}^+$ ($n = 2$–100) close to 958 and 964 nm are plotted in Fig. 3 as a function of the number of helium ad-atoms on the fullerene surface. The absorption wavelength (corrected to vacuum) that was obtained for no helium atoms by Campbell et al.[9] is also indicated, together with their values for up to four attached He atoms (filled symbols). Data for the two weaker $C_{60}^+$ absorption features close to 937 and 943 nm are shown in Supplementary Fig. 3. For a growing number of helium atoms, we observe for the absorption wavelength a remarkably linear red shift of 0.072(1) nm per helium atom until $n = 32$. Beyond $n = 32$, a linear blue shift of 0.046(2) nm per helium atom is observed for the next 12 atoms. At ~60 attached helium atoms we observe a local minimum in the red shift and then, again, a small increase up to 80 helium atoms. For larger clusters up to at least $n = 150$, the absorption wavelength remains constant.

## Discussion

The observed shifts directly reflect the $He_n$-$C_{60}^+$ interaction and shell closure. The first 32 helium atoms occupy the sites above the centres of the hexagonal and pentagonal carbon rings. Each atom has almost the same distance from the chromophore determined by a binding energy of ~9 meV[25] and contributes a similar amount to the red shift of the absorption wavelength with respect to the bare $C_{60}^+$ electronic band position. This yields an almost linear wavelength dependence on the added number of helium atoms until all facets are occupied at 32 attached helium atoms. This complex can be associated with a commensurate decoration where one helium atom is positioned above the centre of each hexagonal and pentagonal face of $C_{60}^+$. Further increase of the

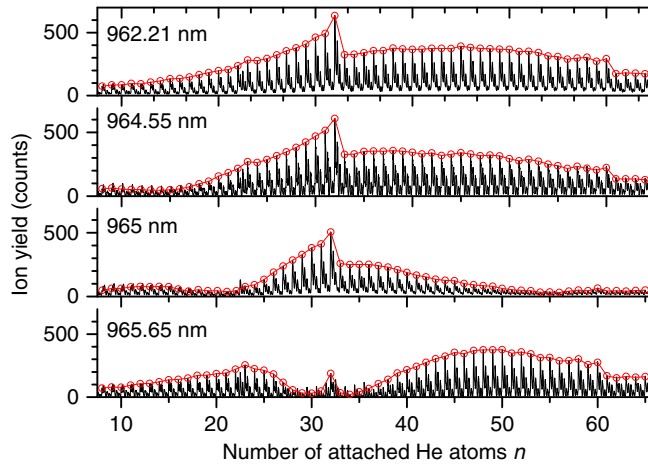

**Figure 1 | Mass spectrum.** Comparison of a mass spectrum where $He_nC_{60}^+$ is transparent (962.21 nm) with mass spectra for three different laser wavelengths at the electronic transition of bare $C_{60}^+$ near 964 nm. Different parts of the mass spectrum are depleted depending on the laser wavelength. The pronounced intensity drops at $n = 32$ and $n = 60$ can be assigned to shell closures of the He adsorbate layer.

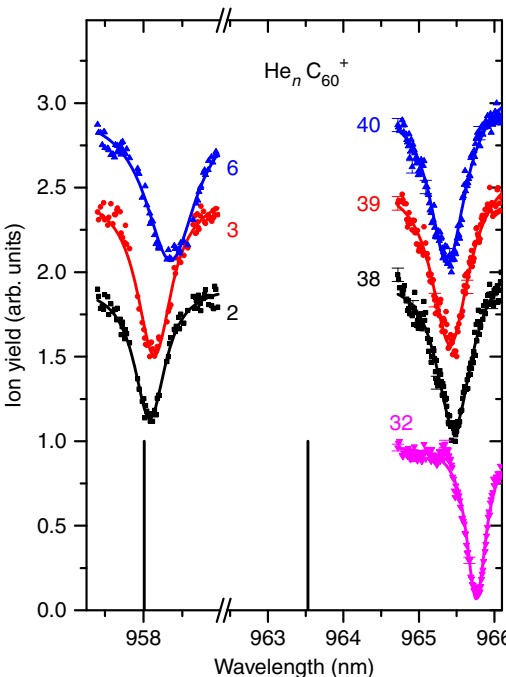

**Figure 2 | Ion signal depletion.** Wavelength scans near 958 and 964 nm for seven different cluster sizes (solid symbols) together with the position of the resonance of the electronic transition for the bare $C_{60}^+$, taken from ref. 9 and corrected from air to vacuum (vertical lines). Error bars indicate the s.d. of the ion yield. Photoabsorption depletes the ion signal to minima at different wavelength positions with a line width of about 0.2 to 0.6 nm (full width at half maximum). The solid lines represent Lorentzian profiles fitted to the data.

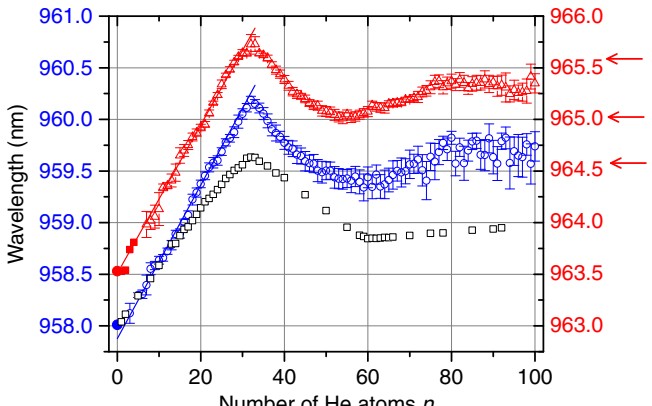

**Figure 3 | Absorption wavelength as a function of He atoms attached.** Centre positions for the absorption spectra of $He_nC_{60}^+$ around 958 nm (blue open circle, left y axis) and 964 nm (red open triangle, right axis) plotted as a function of n, the number of helium ad-atoms on the fullerene ion surface. The error bars indicate s.e.m. of the centre position of the Lorentzian profiles fitted to the ion signal depletion (see Fig. 2). The absorption wavelengths (corrected to vacuum) that were obtained for zero to a few helium atoms by Maier and colleagues[9] are indicated by the bold symbols. The red arrows indicate the wavelengths at which the mass spectra shown in Fig. 1 were measured. The open grey squares represent calculated absorption wavelengths for $He_nC_{60}^+$ including quantum effects, renormalized by a factor of 1.0008.

number of helium atoms results in a sharp kink to smaller absorption wavelengths due to the displacement of helium atoms from the pentagonal central sites. Theoretical studies predict, at this point, the formation of a mobile (liquid) layer intermixed with a solid part (the 20 He atoms that occupy the positions above the centre of the hexagons)[24,25]. A pronounced intensity drop in the mass spectra (see Fig. 1) at around $n = 60$ indicates a shell closure at this cluster size. We note that the red shift for $He_{60}C_{60}^+$ is the same as for $He_{20}C_{60}^+$, thus—in agreement with the theoretical studies[24,25]—indicating that only He atoms positioned above the centre of the 20 hexagonal faces of the fullerene cage are significantly involved in the interaction with $C_{60}^+$. The increased red shift from 60 to 80 attached helium atoms can be attributed to the influence of the outer helium ad-layer. Beyond $n = 80$, we observe an almost constant resonance wavelength. It is possible that this constant absorption wavelength coincides with the onset of superfluidity.

To support the above picture, we have performed elaborate MD simulations by using the realistic force field introduced in ref. 25 and including quantum effects. By combining the results of these simulations with a simple model that describes the van der Waals (vdW) interaction between $C_{60}^+$ and the surrounding helium atoms, we have evaluated the energy shifts in the ground and first excited states of $C_{60}^+$ as a function of the number of helium atoms. The calculated line shifts are also included in Fig. 3 (open squares). The theoretical results show an impressive qualitative similarity to the experiments, with an initial linear increase in the peak wavelength, culminating in a peak in this red shift at $n = 32$, followed by a gradual decrease and bottoming out around $n = 60$, before slightly increasing again. The Supplementary Fig. 4 shows calculated line shifts including and neglecting quantum effects in comparison with the experimental data. Without quantum effects,

the local minimum in the line shift is located around $n = 70$, whereas calculations including quantum effects and the experiment reveal such a minimum at $n = 60$. Furthermore, also the slope of the blue shift from $n = 32$ to this minimum agrees better with the experiment when including quantum effects.

The trends in the shifts of absorption wavelength with increasing number of helium atoms is reminiscent of the vibrational bandshifts measured by McKellar and colleagues[27,28] in neutral molecule—$He_n$ clusters, which were also interpreted in terms of solvation shell closures. As in the present case, the shifts clearly depend on the helium–dopant interactions.

The weaker lines close to the 937 and 943 nm $C_{60}^+$ electronic absorption bands show similar wavelength shifts for helium physisorption (see Supplementary Fig. 3). Extrapolation of all spectra to bare $C_{60}^+$ yields the line positions of $936.74 \pm 0.01$, $943.02 \pm 0.02$, $957.91 \pm 0.016$ and $963.52 \pm 0.03$ nm, in agreement with the data from Campbell et al.[9,16] once corrected to vacuum, since all present data are obtained in vacuum.

The spectroscopic investigation of an Atkins snowball provides unprecedented details about the solvation of ions by helium. By choosing ions with different corrugation and curvature, the balance between surface and mutual interaction can be systematically varied and its effect on the helium phase transitions studied in detail. The theoretical procedure applied in this study is a suitable routine for prescreening the effect of messenger helium atoms to various molecules prior their experimental investigations. In addition, our study demonstrates that doped helium nanodroplets provide a powerful tool to study the spectroscopic characteristics of cations or transient species under isolated and cold conditions relevant to outer space. The weak matrix effect of helium has been used by Maier and colleagues[9] to convincingly attribute two DIBs to the cation $C_{60}^+$ and predict—and recently tentatively confirmed—three additional DIB transitions[15,16]. The linear wavelength shift in our study demonstrates that the absorption wavelength of species isolated in He droplets can be accurately predicted to better than 0.05 nm through extrapolation, and in the wavelength domain of the $C_{60}^+$ bands investigated here, this is comparable to the accuracy

of astronomical observations. Therefore, this technique provides a convenient way to systematically investigate the absorption spectra of astrophysically relevant species, including smaller fullerenes, polycyclic aromatic hydrocarbons and their derivatives that were shown to be linked to fullerene formation in space[29], and may be possible carriers of the other unidentified DIBs.

## Methods

**Experimental.** We have prepared fullerene–helium ion complexes by loading $C_{60}$ molecules into helium nanodroplets and subsequent ionization via electron impact. The resulting ions are analysed by a high-resolution reflectron, time-of-flight mass spectrometer (Tofwerk AG, model HTOF). The details of this experiment have been described previously[30]. The high resolution of the mass spectrometer is used to assign the fullerene–helium ion clusters to their specific atomic composition, because isobaric mass differences of clusters with nominally the same mass are easily resolved. For example, pure $He_{(180+n)}^+$ clusters are distinguished from $He_nC_{60}^+$ due to their mass difference of 0.469 u. After ionization and before detection in the mass spectrometer, the cluster ions are subjected to the radiation of a continuous-wave titanium sapphire (Ti:Sa) laser (Sirah Matisse TR, 10 MHz bandwidth and 0.6 W power). If the Ti:Sa laser hits an electronic transition in the $C_{60}^+$ core, the photo-absorption followed by radiative or non-radiative decay heats up the ion. This triggers evaporation of the weakly bound helium atoms, which is detected as a depletion of the respective cluster signal. In the experiment, mass spectra of all helium cluster ions are taken at the same time, whereas the laser frequency is scanned. An animated sequence of a section of the mass spectra from $m/z = 730$ until $m/z = 990$ as a function of the laser wavelength is shown in the Supplementary Movie 1. This new technique allows for an efficient parallel recording of the absorption spectra of all cluster ions simultaneously and enables systematic studies on the effects of size on the interaction of helium adsorbents.

**General theoretical model.** The vdW interaction between helium atoms and the $C_{60}^+$ cation comes from two parts: (i) the charge/induced dipole interaction between the $C_{60}^+$ cation and the helium atoms, and (ii) the London dispersion interaction between helium atoms and the cage. As a simple approximation, we consider the highly symmetrical $C_{60}^+$ cage as an isotropic sphere. Then, the vdW interaction energy between a single He atom and the $C_{60}^+$ cage can be estimated as[31–33]

$$E_{int} = E_{ind} + E_{disp} = -\frac{e^2\alpha_{He}}{2(4\pi\varepsilon_0)^2R^4} - \frac{3}{2}\frac{IP_{He}IP_{C_{60}^+}}{IP_{He}+IP_{C_{60}^+}}\frac{\alpha_{He}\alpha}{(4\pi\varepsilon_0)^2R^6} \quad (1)$$

where $R$ is the distance of the He atom to the centre of the $C_{60}^+$ cage, $\alpha_{He}$ and $\alpha$ are the polarizabilities of He and $C_{60}^+$, respectively, and $IP_{He}$ and $IP_{C_{60}^+}$ are the first ionization potentials of He and $C_{60}^+$, respectively. As can be seen, $E_{ind}$ only depends on the polarizability of the helium atom and the distance between the latter and the cage centre. Therefore, $E_{ind}$ is expected to be the same for the ground and the first excited state of $C_{60}^+$. The dispersion interaction $E_{disp}$ depends on the polarizability of $C_{60}^+$ (as we will see below), which can change noticeably on electronic transition, thus leading to different shifts on the energy levels of the ground and excited states of $C_{60}^+$. Therefore, here we only need to consider the dispersion interaction to account for the line shifts observed in the experiments.

Hence, the variation of the energy difference between the first excited state $|1\rangle$ and the ground state $|0\rangle$ of $He_nC_{60}^+$ (that is, the line shift) will be given by

$$h\Delta\nu = E_{int}^1 - E_{int}^0 = -\frac{3}{2}\frac{IP_{He}IP_{C_{60}^+}}{IP_{He}+IP_{C_{60}^+}}\frac{\alpha_{He}}{(4\pi\varepsilon_0)^2}\left(\sum_{i=1}^n\frac{1}{R_i^6}\right)\Delta\alpha \quad (2)$$

where $R_i$ is the distance between the $i$-th helium atom and the cage centre, and $\Delta\alpha = \alpha_1 - \alpha_0$ is the difference of polarizability between the ground and the first excited state of $C_{60}^+$. In equation (2), we can use the experimental values: $\alpha_{He} = 1.383746$ a.u.[34,35], $IP_{He} = 24.59$ eV[36] and $IP_{C_{60}^+} = 11.59$ eV[37]. The distances $R_i$ have been obtained from the MD simulations described below (see section MD simulations). To calculate $\Delta\alpha$, which, as we will show below, is of the order of a few tens of atomic units, ideally one should perform high-level *ab initio* calculations. However, these are not feasible for such large systems. Instead, we have used a straightforward model that allows us to estimate the magnitude of $\Delta\alpha$.

**Particle-on-a-sphere model to estimate $\Delta\alpha$.** For the polarizability of aromatic molecules such as $C_{60}$, it has been pointed out that $\sigma$-orbitals give the same contribution to the excited state as to the ground state[38]. Therefore, we will only consider $\pi$-orbitals to calculate $\Delta\alpha$. The simplest way to describe the $\pi$ electrons of $C_{60}^+$ is by using the particle-on-a-sphere model, that is, a particle confined to the 2D surface of a sphere, which is known to have exact solutions of the Schrödinger equation[39]. The eigenfunctions are spherical harmonics $Y_l^m(\theta,\varphi)(l=0,1,2,...; m=0,\pm1,...,\pm l)$ and the eigenenergies are given by

$$E_{l,m} = \frac{\hbar^2}{2m_eR_s^2}l(l+1) = \frac{e^2a_0}{(4\pi\varepsilon_0)2R_s^2}l(l+1) \quad (3)$$

where $R_s$ is the radius of the sphere and $a_0$ the Bohr radius.

The wave function of the ground state of $C_{60}^+$ can be written as a $59\times59$ Slater determinant:

$$\Psi = \left|Y_0^0\overline{Y_0^0}\ldots Y_5^2\overline{Y_5^2}Y_5^{-2}\overline{Y_5^{-2}}Y_5^1\overline{Y_5^1}Y_5^{-1}\overline{Y_5^{-1}}Y_5^0\right| \quad (4)$$

$$\equiv |\ldots 5_2\overline{5_2}5_{-2}\overline{5_{-2}}5_1\overline{5_1}5_{-1}\overline{5_{-1}}5_0| \quad (5)$$

In the last abbreviated notation, only unpaired electrons and the highest occupied molecular orbitals are indicated.

According to perturbation theory, polarizability only comes from the second-order interaction energy, since the first-order perturbation is vanishing due to the odd parity of the dipole operator. The polarizability tensor of the ground state $C_{60}^+$ is then calculated as

$$\alpha_0^{\nu\nu} = 2\sum_i\frac{\langle\Psi_0|\mu_\nu|\Psi_i\rangle\langle\Psi_i|\mu_\nu|\Psi_0\rangle}{E_i - E_0} \quad (6)$$

As a result of spherical symmetry, the polarizability tensor is isotropic:

$$\alpha_0 \equiv \alpha_0^{\nu\nu} = \alpha_0^{zz} = 2e^2\sum_{i>0}\frac{|\langle\Psi_0|\mu_z|\Psi_i\rangle|^2}{E_i - E_0} \quad (7)$$

By using the above expression, it is easy to evaluate the polarizabilities for the ground and first excited states of $C_{60}^+$ (see Supplementary Methods, in particular Supplementary Equations 14 and 28). Hence, the polarizability difference between the first excited and the ground state of $C_{60}^+$ is

$$\Delta\alpha = 2e^2\left[\frac{(3_0|4_0)^2}{E_4 - E_3} + \frac{(5_0|6_0)^2}{E_6 - E_5} - 2\frac{(4_0|5_0)^2}{E_5 - E_4}\right] \quad (8)$$

which from equation (3) and (Supplementary Methods 2) can be written as

$$\Delta\alpha = \frac{80}{9,009}\cdot\frac{4\pi\varepsilon_0R_s^4}{a_0} \quad (9)$$

By using the actual value of the $C_{60}$ radius, which is 6.7 $a_0$ (refs 40,41) equivalent to the distances between the geometrical centre of the cage and carbon atoms, the estimated value of $\Delta\alpha$ is 18 a.u.

**Molecular dynamics simulations.** To determine the values of the $R_i$ distances required in equation (2), we have performed MD simulations for the $He_nC_{60}^+$ systems by using the DL_POLY2 code[42] and the force field introduced in ref. 25. This force field, obtained by fitting a large set of density functional theory (DFT) and coupled-cluster calculations (on the CCSD(T) level), was successfully used in ref. 25 to explain the multi-shell structure of $He_nC_{60}^+$ observed in the experiments.

Quantum effects and dispersion corrections have also been included. The dispersion correction to the DFT energy is included for helium–carbon interactions, following Grimme's DFT-D2 scheme[43]:

$$E_{disp} = -\frac{s_6}{2}\sum_{i,j}\frac{C_6^{ij}}{r_{ij}}f_{damp}(r_{ij}) \quad (10)$$

where the summation is over all pairs of atoms $i$ and $j$; $C_6^{ij}$ is the dispersion coefficient for atom pairs $i$ and $j$; $s_6$ is a scaling factor depending on the functional, $r_{ij}$ is the distance between atoms $i$ and $j$, and $f_{damp}(r_{ij})$ is a damping function so that the dispersion correction takes effects only for long-range interactions. In our simulations, instead of using the recommended values[43], we have calibrated parameters $s_6$ and $C_6^{ij}$, on the basis of high-level *ab initio* calculations (MP2 with complete basis set) for He-pyracylene$^+$ systems.

The quantum effects have been taken into account using Feynman–Hibbs model, where the distribution of quantum particles is approximated by using a Gaussian packet. Accordingly, we have generated an effective Feynman–Hibbs potential from our *ab initio* potential $U(r)$ to second order in $\hbar$[44,45]:

$$U_{FH}(r) = U(r) + \frac{\hbar^2}{24\mu k_BT}\left[U''(r) + 2\frac{U'(r)}{r}\right] \quad (11)$$

where $\mu$ is the reduced mass of the two particles, $k_B$ the Boltzmann constant and $T$ the simulation temperature.

We have tried different initial configurations, including low- and high-energy ones, to guarantee a meaningful statistical sampling of the system. In all calculations, the system was initially heated to 10 K and then slowly cooled at a rate of 0.1 K per 50 ps. Once the system reached 4 K, which is close to the estimated temperature of $He_nC_{60}^+$ under our experimental conditions, the simulation was run for another 5 ns, to ensure full equilibration and to collect statistical information. In all cases, periodic boundary conditions were applied to prevent escape of He atoms.

With all the ingredients at hand, we are now able to predict the line shifts observed in the absorption spectra of $He_nC_{60}^+$, by substituting in equation (2) the calculated values of $\Delta\alpha$ (section 2) and $\sum_{i=1}^n\frac{1}{R_i^6}$ (section 3). In the latter case, the summation is averaged over the last 2ns of the simulation. We have also used a cutoff radius of 6.7 Å to exclude He atoms from the second and further shells[25], because, as indicated by the experimentally measured line shifts and widths, the He-$C_{60}^+$ interaction is much more effective for He atoms belonging to the first shell.

**Data availability.** The data that support the findings of this study are available from the corresponding author upon request.

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

## Acknowledgements

This study was supported by the Fonds zur Förderung der wissenschaftlichen Forschung (FWF) projects P26635, W1259 and I978-N20, Deutsche Forschungsgemeinschaft (DFG) project I978-N20 the European COST Action CM1204 XLIC and the European Research Council under ERC Grant Agreement Number 279898. We acknowledge allocation of computer time at the Centro de Computación Científica of the Universidad Autónoma de Madrid (CCC-UAM) and the Red Española de Supercomputación. Y.W., M.A. and F.M. thank the MINECO projects FIS2013-42002-R and CTQ2013-43698-P, and the CAM project NANOFRONTMAG-CM ref. S2013/MIT-2850 for support.

## Author contributions

A.L., R.W. and P.S. conceived the project. M.K., S.S., M.S., M.R., J.P and S.R. carried out the bulk of the experimental work. Y.W., M.A. and F.M. carried out theoretical calculations. H.L., A.G.G.M.T., J.C., M.K.B., A.M., A.L and P.S. wrote the manuscript. P.S. oversaw all the works. All the authors discussed the results and commented on the manuscript.

## Additional information

**Competing financial interests:** The authors declare no competing financial interests.

