## [Peer Review File · Nature Communications]

Reviewers' comments:

Reviewer #1 (Remarks to the Author):

A. Key results

This work reports very interesting action spectroscopy measurements on C₆₀ cations coated with helium atoms in precisely controlled numbers. The electronic spectra inferred from mass abundances show non-monotonic variations with the number of helium atoms, which based on earlier measurements and calculations provide experimental evidence of various types of 'phases', including rigidlike arrangements up to 32 atoms, single but liquid layer above this size, and the formation of a second superfluid layer above 60 atoms. These non-monotonic size effects, though not unsurprising for finite-size systems, are here manifested with rather good accuracy. Additional calculations are employed to support part of the measurements at low sizes.

B. Originality and interest

The technique of action spectroscopy is not new per se, but here it is used to probe the variations of the physical properties as a function of the number of the messenger atoms, from the variations of the spectra themselves. While this approach may not work for arbitrary systems, it is here used just appropriately.

C. Data & Methodology

As far as purely experimental measurements are concerned (Fig. 1-4), the approach is definitely valid, and the data shown quite convincing. I would be more critical against the theoretical contribution, which only covers the small size regime where the data show linear behavior, and where the robustness of some parameters entering the modeling may not be ensured. The use of classical MD at 1 K for determining the equilibrium distances between the fullerene and the helium atoms is probably highly questionable as well, since for such problems the classical and quantum mechanical structures are really different [see e.g. J. Chem. Phys. 143, 224306 (2015)]. For instance, I would not be surprised that the radii entering Eq. (2) thus deviate by a fraction of angstrom as well.

The presentation is of a good quality overall, although I found the 'movie' showing the evolution of mass spectra somewhat of a gadget.

D. Appropriate use of statistics and treatment of uncertainties

Experimental data show uncertainties in a convincing way, especially in Fig. 4 where they are probably more essential (they would have been a welcome addition in Fig 3 as well). It is unclear where the uncertainties in the calculated lineshifts (supplementary figure 3) originate from. The MD ingredients? The 'tweaking' mentioned by the authors?

E. Conclusions, robustness, validity and reliability

The conclusions are probably supported by the data, except for the limited information carried by the present additional calculations which do not cover the interesting range extending beyond 32 atoms, and where the second shell arises and develops. In particular, the signature of superfluidity (mentioned by the authors) is a bit unclear to me. The experimental data are much more reliable and less free of interpretation or 'tweaking'.

F. Suggested improvements, data for possible revision

The improvements I see would be a more stringent evaluation of the uncertainties in the calculations, in particular a discussion of the limits of the modeling employed. This is of importance since the main paper does not show them. If the calculations could tell something near the sizes where the actual phase transitions take place the entire study would be so much stronger. The vicinities of 32 and 60 helium atoms, in particular, would be worth pursuing further, may they need a more rigorous treatment of the vibrational dynamics.

On a more minor note, I understand the possible astrophysical implications of coating C60+ (which have now been observed in the ISM) with helium (which is an abundant element), but for the general reader it would be valuable to elaborate on the possible presence of helium-coated fullerenes, as the present investigation focuses on.

G. References

Are overall appropriate. Some possible confusion in the theory section, where it is unclear which of the references 23 or 24 provided the potential used in the MD simulations,

H. Clarity and context

The abstract, introduction and conclusions are well written and convincing, except maybe for criticism raised in Sec. F about the connection with astrochemistry and the DIBs. If the authors mean that their experimental device can reach accuracy that is sufficient for discussing astrophysical models based on specific molecules prone to mass spectrometry measurements, then probably it should be stated more clearly.

Reviewer #2 (Remarks to the Author):

The manuscript reports on the spectroscopy of C60+ dressed with helium atoms. The spectra show some intriguing shift with the number of helium atoms. Up to 32 helium atoms, a nearly constant red shift of 0.07 nm per helium atom is observed, while in the range from 32-60 helium atoms a blue shift of 0.04 nm/atom is observed. For even larger number of helium atoms again a red shift is observed. Following the theoretical prediction of Calvo [ref. 23] based on path-integral methods, these observations are attributed to phase transitions as the number of helium number of atoms changes. Initially the helium atoms are strongly bound to the centers of the 32 hexagonal and

pentagonal rings, corresponding to solid structure. Addition of up to 60 helium atoms increases the delocalization energy and gives rise to a liquid like structure, while addition of even more atoms leads again to a solid like structure. The simulation of the wavelength shifts based on classical molecular dynamics reported in the manuscript appears to be in agreement with this picture, thereby providing evidence for this interesting phenomenon.

While the authors report on a very intriguing and scientifically interesting observation, I feel they are trying to oversell their work. It starts with the title, which claim that the C60+ ions are solvated in superfluid helium droplets. From experiments and also theory it is absolutely not obvious that the systems are superfluid. As a matter of fact, the authors report an estimated temperature of 4 K the C60+He_n system, well above the helium superfluid temperature of 2 K. Later in the analysis of the data, line 94, they attribute the constant resonance wavelength for more than 80 helium atoms with the onset of superfluidity. This is mere speculation, they have absolutely no evidence for this claim. Actually, in the supporting information they state themselves that helium atoms in the second solvation layer, which are those having more than 72 atoms, do not contribute to measurable wavelength shift.

Linked to this, relevant works related to (superfluid) phase transition in helium clusters containing neutral dopants or spectroscopy of ions in helium droplets are not referenced.

Lastly the claim that the accuracy by which the transition wavelength of the bare ion can be determined by extrapolating the shifts to zero helium atoms to be relevant for astronomical identification is questionable. The extrapolated values reported in line 113 have uncertainties of ~0.02 nm but deviate systematically by approximately 0.1 nm from the known values, which significantly more than the claimed accuracy of 0.05 nm reported in line 130. Furthermore, it is not obvious that less symmetrical molecules provide such a systematic variation of the wavelength shift as observed for C60+, as is evidenced by the work Even on neutral species [ref. 10].

In addition, there are a few other issues the authors should address.

Line 48: "...characteristic wavelength shift (0.02 nm for one adsorbed He atom)¹⁰ introduced into electronic transitions by the He-C60+ interaction." This statement is misleading as it refers to the interaction of helium with neutral species, and later a much larger value is reported for He-C60+.

Line 63, Figure 2: The authors should only report their own spectra. A direct comparison with the spectrum of the bare ion is unnecessary and might give the impression that they have also measured this, which is not the case.

Line 66: "...can be explained by isomeric broadening (see below)." This is nowhere explained in detail later on in the text. Related to this the authors should also discuss the effect of rovibronic structure in these spectra on the linewidth. As it is now Figure 4 provides no relevant information and might just as well be removed.

Line 107: "The reduced red-shift of the calculated resonance wavelengths between $n = 20$ and 32 , as well as the small down-deviation of the experimental data from the linear trend in the same interval, suggest that the sites associated with the hexagons are the preferred adsorption sites and will be filled first." This sentence poses a problem. If I look at figure 3, I do not see such a downshift as the text implies. If it were there then the question arises why have the authors included it in the fit to determine the transition wavelength of the bare ion, as the straight line seems to suggest.

Line 124-129: This section on the diffuse interstellar bands is not relevant for the current work, which discusses phase transitions in helium on C60+ and should be removed from the manuscript.

Concluding, the manuscript describes a remarkable observation related to predicted phase transitions in helium on the surface of C₆₀⁺ that might appeal to a broad audience and thereby is suitable for publication in Nature Communications. However, the authors at times claim more than the data corroborate. In its present form the manuscript should therefore not be published.

Reviewer #3 (Remarks to the Author):

The authors describe a study of successive solvation of a C₆₀⁺ ion with helium atoms. A new technique was developed, in which neutral C₆₀ is first embedded into helium nanodroplets and then ionized. The resulting C₆₀He_n⁺ fragments are then irradiated with a Ti:Sa laser, which is followed by detection with a time-of-flight mass spectrometer. Upon excitation, helium atoms are evaporated and the helium atom loss is recorded as a reduction in ion count as function of laser wavelength. The authors were able to identify specific details in the development of the solvation shell structure, such as occupation of specific sites by helium atoms and solvation shell formation. The authors also identify different phases of the helium coating, i.e. solid, liquid, and superfluid phases. The experimental results are supported by, and interpreted with the help of, molecular dynamics simulations.

I feel that these are important results, which warrant publication in Nature Communications. This work supports the recent spectacular assignment of several diffuse interstellar bands to C₆₀⁺ by Meier and coworkers through extrapolation of the absorption wavelengths of C₆₀He_n⁺. I believe that the authors are correct in their claim that the technique used is more general and can be applied to other ionic species. Furthermore, the observed wavelength shifts of the absorption features provide important insights into the development and the structure of the helium solvation shell. I have a number of comments (some very minor) which, I think, the authors should consider before the manuscript is accepted.

1) line 53: "Figure 1 shows one mass spectrum off-resonance with the ..." better "Figure 1 shows one mass spectrum off-resonant with the ..." ?

2) line 84: "... a 1×1 commensurate decoration ..." perhaps it's better to say '1 to 1' instead of '1 by 1'?

3) line 89: "... positions on-top the center of the hexagons ..." better "... positions above the centers of the hexagons ..." ?

4) lines 94 - 96: The authors interpret the constant absorption wavelength above n=80 in terms of superfluidity of the outer helium layer. However, it seems to me that there is no evidence for this interpretation. For example, I don't believe that the band shifts can be interpreted in terms of a superfluid response using linear response theory, as has been done for dopant molecule rotation. I think the authors should use a more cautious wording, for example, ... we speculate that the constant absorption wavelength coincides with the onset of superfluidity, but more rigorous simulations are needed to confirm this ...

5) The trends in the shifts of absorption wavelength with increasing number of helium atoms is reminiscent of the vibrational bandshifts measured by McKellar in neutral molecule - He_n clusters, which were also interpreted in terms of solvation shell closures. In both cases, the shifts clearly depend on the helium - dopant interactions, and perhaps it's worthwhile to point this out in the paper.

6) line 122: I'm not sure if 'This property ...' refers to helium nanodroplet experiments. Meier and

coworkers have not used helium nanodroplets in their work.

7) Figure 2: It seems to me that the color coding and symbol coding is unnecessary.

8) Figure 3: Either the color coding or the symbol coding is unnecessary. The meaning of the solid lines, which envelope the data points, need to be explained.

9) On line 66, the authors promise that an explanation of the increased linewidths in terms of isomeric broadening will be given later in the manuscript. However, the linewidths are not referred to again. I feel that Figure 4 can simple be omitted.

10) Supplementary Information: "After ionization and before detection in the mass spectra the cluster ions are ..." should it read 'mass spectrometer'?

11) It would be nice to have complete spectra (as function of wavelength, similar to those in Figure 2, but for the complete wavelength range) for several n-values in the Supplementary Material. These could be presented in a stacked form, and would convey better (at least to me) how the absorption features shift with increasing n. (Maybe the wavelength scale is too large to see the small shifts?)

Reviewer #1:

A. Key results

This work reports very interesting action spectroscopy measurements on C₆₀ cations coated with helium atoms in precisely controlled numbers. The electronic spectra inferred from mass abundances show non-monotonic variations with the number of helium atoms, which based on earlier measurements and calculations provide experimental evidence of various types of 'phases', including rigidlike arrangements up to 32 atoms, single but liquid layer above this size, and the formation of a second superfluid layer above 60 atoms. These non-monotonic size effects, though not unsurprising for finite-size systems, are here manifested with rather good accuracy. Additional calculations are employed to support part of the measurements at low sizes.

B. Originality and interest

The technique of action spectroscopy is not new per se, but here it is used to probe the variations of the physical properties as a function of the number of the messenger atoms, from the variations of the spectra themselves. While this approach may not work for arbitrary systems, it is here used just appropriately.

So far we have observed for all ions attachment of He atoms. But the reviewer is right that the yield of these He tagged ions depends strongly on the nature of the ion. In some cases the intensity is not sufficiently high to perform action spectroscopy measurements with the present setup. Recent results on coronene (Kurzthaler et al. JCP 145 (2016) 064305; DOI: 10.1063/1.4960611) indicate that poly-cyclic aromatic hydrocarbons are perfectly suitable molecules that are also interesting candidates for the interstellar medium.

C. Data & Methodology

As far as purely experimental measurements are concerned (Fig. 1-4), the approach is definitely valid, and the data shown quite convincing. I would be more critical against the theoretical contribution, which only covers the small size regime where the data show linear behavior, and where the robustness of some parameters entering the modeling may not be ensured. The use of classical MD at 1 K for determining the equilibrium distances between the fullerene and the helium atoms is probably highly questionable as well, since for such problems the classical and quantum mechanical structures are really different [see e.g. J. Chem. Phys. 143, 224306 (2015)]. For instance, I would not be surprised that the radii entering Eq. (2) thus deviate by a fraction of angstrom as well.

The size regime has been extended according to the suggestion of the reviewer, particularly in the vicinity of the sizes where phase transitions are expected, i.e., 32 and 60 He

atoms. Furthermore, we included quantum effects in our simulations with only minor deviations to the classical calculations. A comparison between classical MD and MD including quantum effects is shown in Figure 5 of the Supplementary Material.

The presentation is of a good quality overall, although I found the 'movie' showing the evolution of mass spectra somewhat of a gadget.

We include this movie as it shows nicely how the photons of a specific energy interact with ions having a particular number of He atoms attached.

D. Appropriate use of statistics and treatment of uncertainties

Experimental data show uncertainties in a convincing way, especially in Fig. 4 where they are probably more essential (they would have been a welcome addition in Fig 3 as well). It is unclear where the uncertainties in the calculated lineshifts (supplementary figure 3) originate from. The MD ingredients? The 'tweaking' mentioned by the authors?

The error bars in Figure 3 were indicated by two solid lines. In the revised version we plotted them in a more common way.

Concerning the uncertainties of the calculations, we expect that the arrangement of the He atoms has a subtle influence on the outcome which will result in a broadening of the line-shift. However, as we removed Figure 4 from the manuscript as suggested by reviewer 3 we omit this discussion here.

E. Conclusions, robustness, validity and reliability

The conclusions are probably supported by the data, except for the limited information carried by the present additional calculations which do not cover the interesting range extending beyond 32 atoms, and where the second shell arises and develops. In particular, the signature of superfluidity (mentioned by the authors) is a bit unclear to me. The experimental data are much more reliable and less free of interpretation or 'tweaking'.

Following the suggestion of the reviewer, we performed additional calculations that cover the entire experimental range of cluster sizes (see below). Concerning the signature of superfluidity the referee is right and we changed this section accordingly to: We speculate that the constant absorption wavelength coincides with the onset of superfluidity, but more rigorous simulations are needed to confirm this.

F. Suggested improvements, data for possible revision

The improvements I see would be a more stringent evaluation of the uncertainties in the calculations, in particular a discussion of the limits of the modeling employed. This is of importance since the main paper does not show them. If the calculations could tell some-

thing near the sizes where the actual phase transitions take place the entire study would be so much stronger. The vicinities of 32 and 60 helium atoms, in particular, would be worth pursuing further, may they need a more rigorous treatment of the vibrational dynamics.

We thank the referee for the valuable comments and suggestions to improve our calculations. We performed new calculations including quantum effects and the agreement between experiment and calculations substantially improved. Whereas Figure 3 in the manuscript only shows the results of the calculations including quantum effects we show a comparison between experimental results and calculations with and without quantum effects in Figure 4 of the Supplementary Materials. Appropriate changes have been made to the text. As suggested by the reviewer we calculated all clusters in the vicinities of 32 and 60 helium atoms.

On a more minor note, I understand the possible astrophysical implications of coating C60+ (which have now been observed in the ISM) with helium (which is an abundant element), but for the general reader it would be valuable to elaborate on the possible presence of helium-coated fullerenes, as the present investigation focuses on.

We do not expect that He covered fullerene ions are present in the ISM. However, as demonstrated in many spectroscopic studies He is the perfect messenger as it provides the weakest matrix effect and therefore the absorption lines of the bare ions can be determined. The present method can easily be utilized for other ions with relevance for the ISM such as poly-cyclic aromatic hydro carbons.

G. References

Are overall appropriate. Some possible confusion in the theory section, where it is unclear which of the references 23 or 24 provided the potential used in the MD simulations,

We thank the reviewer in pointing out a mistake in the numbering of the references. On page 4 of the original manuscript we wrote 22,23 which should have been 23,24. Due to the inclusion of the new reference 16 it has been changed to 24,25 in the revised manuscript.

H. Clarity and context

The abstract, introduction and conclusions are well written and convincing, except maybe for criticism raised in Sec. F about the connection with astrochemistry and the DIBs. If the authors mean that their experimental device can reach accuracy that is sufficient for discussing astrophysical models based on specific molecules prone to mass spectrometry measurements, then probably it should be stated more clearly.

This study is the first independent laboratory based proof of the data presented by Campbell et al. Given the importance of their findings, an independent study confirming their data will further strengthen the case that after nearly 100 years the first DIBs features now have been assigned. However, so far only C_3 and C_{60}^+ have been identified in translucent clouds and there are some 400+ DIBs left for assignment. The present methodology has the full potential to search for these carriers and whereas indeed a focus in this work is by showing for a first time a high resolution study monitoring the solvation dynamics of a large molecular ion, it also shows the potential of the present method to search for the electronic transitions of species chemically related to fullerenes. This information has now been added in the text.

Reviewer #2:

The manuscript reports on the spectroscopy of C_{60}^+ dressed with helium atoms. The spectra show some intriguing shift with the number of helium atoms. Up to 32 helium atoms, a nearly constant red shift of 0.07 nm per helium atom is observed, while in the range from 32-60 helium atoms a blue shift of 0.04 nm/atom is observed. For even larger number of helium atoms again a red shift is observed. Following the theoretical prediction of Calvo [ref. 23] based on path-integral methods, these observations are attributed to phase transitions as the number of helium number of atoms changes. Initially the helium atoms are strongly bound to the centers of the 32 hexagonal and pentagonal rings, corresponding to solid structure. Addition of up to 60 helium atoms increases the delocalization energy and gives rise to a liquid like structure, while addition of even more atoms leads again to a solid like structure. The simulation of the wavelength shifts based on classical molecular dynamics reported in the manuscript appears to be in agreement with this picture, thereby providing evidence for this interesting phenomenon.

While the authors report on a very intriguing and scientifically interesting observation, I feel they are trying to oversell their work. It starts with the title, which claims that the C_{60}^+ ions are solvated in superfluid helium droplets. From experiments and also theory it is absolutely not obvious that the systems are superfluid. As a matter of fact, the authors report an estimated temperature of 4 K the C_{60}^+ -He system, well above the helium superfluid temperature of 2 K. Later in the analysis of the data, line 94, they attribute the constant resonance wavelength for more than 80 helium atoms with the onset of superfluidity. This is mere speculation, they have absolutely no evidence for this claim. Actually, in the supporting information they state themselves that helium atoms in the second solvation layer, which are those having more than 72 atoms, do not contribute to measurable wavelength shift.

We appreciate the concerns of the reviewer, but we are confident about the fact that from 80 He-atoms onwards the C_{60}^+ is solvated in a superfluid droplet.

The temperature of 4K in our PRL paper from 2012 was estimated from the binding energy of 10meV for up to n=32 He atoms attached. However, for n=80 the binding energy of an additional He atom drops to basically zero (see Figure 2 in the work of Calvo, PRB 2012). Thus the temperature of a stable $C_{60}^+He_{80}$ can be expected to be way below 2K, fully in line with the argument given by the reviewer, and thus we conclude that the weakly bound He atoms (for 80 atoms or more) are indeed superfluid.

Linked to this, relevant works related to (superfluid) phase transition in helium clusters containing neutral dopants or spectroscopy of ions in helium droplets are not referenced. By adding a section on vibrational bandshifts for neutral molecules relevant studies related to superfluid phase transitions by the groups of Jäger and Mc Kellar have been cited.

Lastly the claim that the accuracy by which the transition wavelength of the bare ion can be determined by extrapolating the shifts to zero helium atoms to be relevant for astronomical identification is questionable. The extrapolated values reported in line 113 have uncertainties of ~0.02 nm but deviate systematically by approximately 0.1 nm from the known values, which significantly more than the claimed accuracy of 0.05 nm reported in line 130. Furthermore, it is not obvious that less symmetrical molecules provide such a systematic variation of the wavelength shift as observed for C_{60}^+ , as is evidenced by the work Even on neutral species [ref. 10].

We assume that the reviewer refers to the data by Campbell et al, when speaking about the 'known' values. So, the argument would only hold when these data are more precise than ours, and we wonder on the basis of which argument this is concluded. Our data set comprises 60 points in the extrapolation, the study by Campbell et al. has 4. We claim a 0.02nm precision which is 10% of the typical FWHM of the involved DIBS. And even with a 0.1nm absolute mismatch (which we doubt) this would still be more than sufficient to compare with the available DIB spectra.

In addition, there are a few other issues the authors should address.

Line 48: "...characteristic wavelength shift (0.02 nm for one adsorbed He atom)¹⁰ introduced into electronic transitions by the He- C_{60}^+ interaction." This statement is misleading as it refers to the interaction of helium with neutral species, and later a much larger value is reported for He- C_{60}^+ .

Both the work of the Maier group as well as our study only refers to the interaction of helium with charged fullerenes.

Line 63, Figure 2: The authors should only report their own spectra. A direct comparison with the spectrum of the bare ion is unnecessary and might give the impression that they have also measured this, which is not the case.

It is not uncommon to include complementary data in a study, as long as these are well cited. We make no claim whatsoever in the paper that we recorded these spectra, but we follow the reviewer in this respect. We have removed the spectra and vertical sticks indicate now the absorption positions. Appropriate changes have been made to the text.

Line 66: "...can be explained by isomeric broadening (see below)." This is nowhere explained in detail later on in the text. Related to this the authors should also discuss the effect of rovibronic structure in these spectra on the linewidth. As it is now Figure 4 provides no relevant information and might just as well be removed.

We agree with the reviewer that in the present form this figure does not provide relevant information and thus removed Figure 4. With additional input from theory we plan to publish this in a forthcoming paper.

Line 107: "The reduced red-shift of the calculated resonance wavelengths between $n = 20$ and 32 , as well as the small down-deviation of the experimental data from the linear trend in the same interval, suggest that the sites associated with the hexagons are the preferred adsorption sites and will be filled first." This sentence poses a problem. If I look at figure 3, I do not see such a downshift as the text implies. If it were there then the question arises why have the authors included it in the fit to determine the transition wavelength of the bare ion, as the straight line seems to suggest.

Calculated red shifts that include quantum effects do not reveal such a pronounced difference between the hexagonal and pentagonal sites. Furthermore, they agree much better with the experiments concerning a local minimum for the redshift at $n=60$. In the revised version of the manuscript we show now these more advanced calculations (see also reviewer #1).

Line 124-129: This section on the diffuse interstellar bands is not relevant for the current work, which discusses phase transitions in helium on C_{60}^+ and should be removed from the manuscript.

We do not agree and for a specific reason. This study is the first independent laboratory based proof of the data presented by Campbell et al. Given the importance of their findings, an independent study confirming their data will further strengthen the case that after nearly 100 years the first DIBs features now have been assigned. However, so far only C_3 and C_{60}^+ have been identified in translucent clouds and there are some 400+ DIBs left for assignment. The present methodology has the full potential to search for these carriers and whereas indeed a focus in this work is by showing for a first time a high resolution study monitoring the solvation dynamics of a large molecular ion, it also shows the potential of the present method to search for the electronic transitions of species chemically related to fullerenes. This information has now been added in the text.

Reviewer #3

The authors describe a study of successive solvation of a C_{60}^+ ion with helium atoms. A new technique was developed, in which neutral C_{60} is first embedded into helium nanodroplets and then ionized. The resulting $C_{60}He_n^+$ fragments are then irradiated with a Ti:Sa laser, which is followed by detection with a time-of-flight mass spectrometer. Upon excitation, helium atoms are evaporated and the helium atom loss is recorded as a reduction in ion count as function of laser wavelength. The authors were able to identify specific details in the development of the solvation shell structure, such as occupation of specific sites by helium atoms and solvation shell formation. The authors also identify different phases of the helium coating, i.e. solid, liquid, and superfluid phases. The experimental results are supported by, and interpreted with the help of, molecular dynamics simulations.

I feel that these are important results, which warrant publication in Nature Communications. This work supports the recent spectacular assignment of several diffuse interstellar bands to C_{60}^+ by Meier and coworkers through extrapolation of the absorption wavelengths of $C_{60}He_n^+$. I believe that the authors are correct in their claim that the technique used is more general and can be applied to other ionic species. Furthermore, the observed wavelength shifts of the absorption features provide important insights into the development and the structure of the helium solvation shell. I have a number of comments (some very minor) which, I think, the authors should consider before the manuscript is accepted.

1) line 53: "Figure 1 shows one mass spectrum off-resonance with the ..." better "Figure 1 shows one mass spectrum off-resonant with the ..." ?

done

2) line 84: "... a 1×1 commensurate decoration ..." perhaps it's better to say '1 to 1' instead of '1 by 1'?

We removed 1×1 and the text now reads: This complex can be associated with a commensurate decoration where one helium atom is positioned above the center of each hexagonal and pentagonal face of C_{60}^+ .

3) line 89: "... positions on-top the center of the hexagons ..." better "... positions above the centers of the hexagons ..." ?

done

4) lines 94 - 96: The authors interpret the constant absorption wavelength above $n=80$ in terms of superfluidity of the outer helium layer. However, it seems to me that there is no evidence for this interpretation. For example, I don't believe that the band shifts can be interpreted in terms of a superfluid response using linear response theory, as has been done for dopant molecule rotation. I think the authors should use a more cautious wording, for example, ... we speculate that the constant absorption wavelength coincides with the onset of superfluidity, but more rigorous simulations are needed to confirm this ...

The referee is right and we changed this section accordingly to: We speculate that the constant absorption wavelength coincides with the onset of superfluidity, but more rigorous simulations are needed to confirm this.

5) The trends in the shifts of absorption wavelength with increasing number of helium atoms is reminiscent of the vibrational bandshifts measured by McKellar in neutral molecule - He_n clusters, which were also interpreted in terms of solvation shell closures. In both cases, the shifts clearly depend on the helium - dopant interactions, and perhaps it's worthwhile to point this out in the paper.

This is a good point that we gladly include in the revised manuscript. We added: The trends in the shifts of absorption wavelength with increasing number of helium atoms is reminiscent of the vibrational bandshifts measured by the groups of Jäger and Mc Kellar in neutral molecule - He_n clusters, which were also interpreted in terms of solvation shell closures^{26,27}. As in the present case, the shifts clearly depend on the helium - dopant interactions.

6) line 122: I'm not sure if 'This property ...' refers to helium nanodroplet experiments. Meier and coworkers have not used helium nanodroplets in their work.

We changed this to: The weak matrix effect of helium has been used by Maier and coworkers ...

7) Figure 2: It seems to me that the color coding and symbol coding is unnecessary.

The filled squares and circles are in the actual size of the figure hard to distinguish and thus color coding will help the reader particularly of an electronic version of the manuscript.

8) Figure 3: Either the color coding or the symbol coding is unnecessary. The meaning of the solid lines, which envelope the data points, need to be explained.

As the y-axis is different for the two data sets color coding is quite helpful. We changed the lines that indicate the uncertainty of the data point to more common error bars (see also reviewer #1).

9) On line 66, the authors promise that an explanation of the increased linewidths in terms of isomeric broadening will be given later in the manuscript. However, the linewidths are not referred to again. I feel that Figure 4 can simple be omitted.

Isomeric broadening is an interesting observation and we thank the reviewer to point out that we deleted the promised explanation from the manuscript. We changed "(see below)." to "The redshift for two or more attached He atoms that are located next to each other is expected to be slightly different, due to the He-He interaction, compared to an isomer where the same number of He atoms are attached widely separated from each other. For a partly filled layer different isomeric species will result in an increased linewidth."

10) Supplementary Information: "After ionization and before detection in the mass spectra the cluster ions are ..." should it read 'mass spectrometer'?

done

11) It would be nice to have complete spectra (as function of wavelength, similar to those in Figure 2, but for the complete wavelength range) for several n-values in the Supplementary Material. These could be presented in a stacked form, and would convey better (at least to me) how the absorption features shift with increasing n. (Maybe the wavelength scale is too large to see the small shifts?)

We added a new Figure 3 to the supplementary materials that shows wavelength scans for a larger number of n-values that cover the whole line-shift range we observe.

REVIEWERS' COMMENTS:

Reviewer #1 (Remarks to the Author):

In this revision the authors have addressed all issues raised in my previous report, performing additional simulations when necessary and providing further justification about the astrophysical relevance of the investigated systems. I still have not fully understood the computational model, which seems to incorporate quite a number of phenomenological ingredients such as the dispersion correction for MD (I would have thought such a contribution is already included in the force field in section 3 of SI).

I noticed a number of typos in the revision, but nothing dramatic.

I now recommend this manuscript for publication in Nature Communications.

Reviewer #2 (Remarks to the Author):

The revised manuscript has been improved in many aspects. However, there are still some issues the authors should address before publication.

As referee 3 and I already mentioned before, there is no evidence in the experiments nor the classical calculations for superfluidity. The authors agree to this and in the revised manuscript carefully phrase that the spectroscopic signature beyond $n=80$ might coincide with the onset of superfluidity (line 99). I fully agree with this formulation. However, in the introduction they claim in line 31 to have observed the transition from liquid to superfluid. The same holds for the formulation in line 50. The authors should also rephrase these sentences more carefully. Along this same line, I strongly oppose to the term superfluidity in the title of the manuscript. As long as there is no direct evidence -an estimated temperature below 2 K cannot be considered as such- its use in the title is not justified.

I am still not convinced by the authors' arguments that the manuscript benefits from the discussion on the DIBs. In my view, it distracts from the main message they want to convey. I suggest reducing this section by eliminating lines 127-132.

Some minor issues:

Line 48: "...characteristic wavelength shift (0.02 nm for one adsorbed He atom)¹⁰ introduced into electronic transitions by the He-C60 + interaction...". As it is phrased right now one might get the impression that the shift for He-C60+ corresponds to 0.02 nm while later in the text an experimental value of 0.07 nm is reported. The authors should rephrase this to remove any ambiguity.

Throughout the text, the authors seemingly arbitrarily use corrected for vacuum and corrected to vacuum. Assuming they always imply the same, I suggest using a single formulation.

Reviewer #3 (Remarks to the Author):

The authors have addressed the reviewers' remarks and concerns adequately. The manuscript can be accepted for publication.

Reviewer #1:

In this revision the authors have addressed all issues raised in my previous report, performing additional simulations when necessary and providing further justification about the astrophysical relevance of the investigated systems. I still have not fully understood the computational model, which seems to incorporate quite a number of phenomenological ingredients such as the dispersion correction for MD (I would have thought such a contribution is already included in the force field in section 3 of SI).

I noticed a number of typos in the revision, but nothing dramatic.

Reviewer #2:

The revised manuscript has been improved in many aspects. However, there are still some issues the authors should address before publication.

As referee 3 and I already mentioned before, there is no evidence in the experiments nor the classical calculations for superfluidity. The authors agree to this and in the revised manuscript carefully phrase that the spectroscopic signature beyond $n=80$ might coincide with the onset of superfluidity (line 99). I fully agree with this formulation. However, in the introduction they claim in line 31 to have observed the transition from liquid to superfluid. The same holds for the formulation in line 50. The authors should also rephrase these sentences more carefully.

The sentences have been rephrased as requested by the reviewer.

Along this same line, I strongly oppose to the term superfluidity in the title of the manuscript. As long as there is no direct evidence -an estimated temperature below 2 K cannot be considered as such- its use in the title is not justified.

The title has been adapted accordingly.

I am still not convinced by the authors' arguments that the manuscript benefits from the discussion on the DIBs. In my view, it distracts from the main message they want to convey. I suggest reducing this section by eliminating lines 127-132.

The lines in question have been removed.

Line 48: "...characteristic wavelength shift (0.02 nm for one adsorbed He atom)¹⁰ introduced into electronic transitions by the He-C60 + interaction...". As it is phrased right now one might get the impression that the shift for He-C60+ corresponds to 0.02 nm while later in the text an experimental value of 0.07 nm is reported. The authors should rephrase this to remove any ambiguity.

We changed the text in brackets to: "0.02 nm for the first adsorbed He atom".

Throughout the text, the authors seemingly arbitrarily use corrected for vacuum and corrected to vacuum. Assuming they always imply the same, I suggest using a single formulation.

The formulations have been unified to "to vacuum".

Reviewer #3:

The authors have addressed the reviewers' remarks and concerns adequately. The manuscript can be accepted for publication.

Additional Changes:

- *) Affiliation #4 has been added.
- *) We added section headings (Abstract, Introduction, Results, Discussion, Methods) according to the requirements of Nature Communications.
- *) In the second paragraph of the discussion section, we moved “an including quantum effects” from the second sentence to the first for clarity.
- *) Later in the same paragraph we added three sentences referring to Supplementary Fig. 4 which shows the differences of the calculation with and without aforementioned quantum effects.
- *) We moved the important parts of the Supplementary Methods section to the main Methods section (as suggested by the editor) and hence also several additional references were transferred.
- *) We added the data availability statement.
- *) References have been formatted according to Nature style.
- *) Some more acknowledgments have been added and slightly rephrased.
- *) The contributions section has been shortened.
- *) Caption of Figure 1 and throughout the manuscript: the naming of the ionic complexes has been unified to He_nC_{60} (as opposed to C_{60}He_n).
- *) Caption of Figure 2: A sentence has been added that explains the meaning of the solid lines in the figure.
- *) Captions of Figures 2 and 3: Meanings of the error bars have been declared.